# *"Take the treatment and be brave"*: Care experiences of pregnant women with rifampicin-resistant tuberculosis

Marian Loveday[1,2]*, Sindisiwe Hlangu[1], Jennifer Furin[3]

**1** HIV Prevention Research Unit, South African Medical Research Council, KwaZulu-Natal, South Africa, **2** CAPRISA-MRC HIV-TB Pathogenesis and Treatment Research Unit, Doris Duke Medical Research Institute, University of KwaZulu-Natal, Durban, South Africa, **3** Department of Global Health and Social Medicine, Harvard Medical School, Boston, Massachusetts, United States of America

* marian.loveday@mrc.ac.za

**Data Availability Statement:** The primary data is not available as open access was not approved by the South African Medical Research Council Human Research Ethics Committee. However, the

## Abstract

### Background

There are few data on the on the care experiences of pregnant women with rifampicin-resistant TB.

### Objective

To describe the treatment journeys of pregnant women with RR-TB—including how their care experiences shape their identities—and identify areas in which tailored interventions are needed.

### Methods

In this qualitative study in-depth interviews were conducted among a convenience sample from a population of pregnant women receiving treatment for RR-TB. This paper follows COREQ guidelines. A thematic network analysis using an inductive approach was performed to analyze the interview transcripts and notes. The analysis was iterative and a coding system developed which focused on the care experiences of the women and how these experiences affected their perceptions of themselves, their children, and the health care system in which treatment was received.

### Results

Seventeen women were interviewed. The women described multiple challenges in their treatment journeys which required them to demonstrate sustained resilience (i.e. to "be brave"). Care experiences required them to negotiate seemingly contradictory identities as both new mothers—"givers of life"—and RR-TB patients facing a complicated and potentially deadly disease. In terms of their "pregnancy identity" and "RR-TB patient identity" that emerged as part of their care experiences, four key themes were identified that appeared to have elements that were contradictory to one another (contradictory areas). These included: 1) the experience of physical symptoms or changes; 2) the experience of the "mothering"

minimal data set is available. The non-author, institutional point of contact who will be able to field data access queries is Ms Ntombifikile Mbatha (email: Ntombifikile.mbatha@mrc.ac.za). To request the minimal data set, the name of the data set is: "Pregnancy_discrimination_qualitative_study_2020." The variables names are qualitative_data_set_2020" and the coding scheme to request would be "identity_codes" and "advice_codes". We have made the in-depth questionnaire available (uploaded as Appendix 1).

**Funding:** This work was supported by the South African Medical Research Council. The funder had no role in study design, data collection and analysis, decision to publish or preparation of the manuscript.

**Competing interests:** The authors have declared that no competing interests exist.

and "patient" roles; 3) the experience of the care they received for their pregnancy and their RR-TB; and 4) the experience of community engagement. There were also three areas that overlapped with both roles and during which identity was negotiated/reinforced and they included: 1) faith; 2) socioeconomic issues; and 3) long-term concerns over the child's health. At times, the health care system exacerbated these challenges as the women were not given the support they needed by health care providers who were ill-informed or angry and treated the women in a discriminatory fashion. Left to negotiate this confusing time period, the women turned to faith, their own mothers, and the fathers of their unborn children.

## Conclusion

The care experiences of the women who participated in this study highlight several gaps in the current health care system that must be better addressed in both TB and perinatal services in order to improve the therapeutic journeys for pregnant women with RR-TB and their children. Suggestions for optimizing care include the provision of integrated services, including specialized counseling as well as training for health care providers; engagement of peer support networks; provision of socioeconomic support; long-term medical care/follow-up for children born to women who were treated for RR-TB; and inclusion of faith-based services in the provision of care.

## Introduction

Pregnant women are a vulnerable population when it comes to tuberculosis (TB) [1], a fact that has been documented in the medical literature since the mid-1940s [2]. Not only are pregnant women at risk for becoming sick with TB but they and their unborn children are also at risk for adverse pregnancy outcomes, unsuccessful TB treatment outcomes, and higher mortality [3]. The recently conducted Tshepiso study from South Africa found that HIV-positive women with active TB disease during pregnancy had a higher risk of delivering low-birth-weight babies, babies with prolonged hospitalization after birth or babies who died compared to women without TB. These women also had higher rates of maternal hospitalization and pre-eclampsia [4]. A variety of factors may be contributing to the development of both TB disease and these poor outcomes, including: immunological changes during pregnancy [5]; a lack of systematic screening for TB among pregnant women [6]; concomitant HIV disease [7]; provider hesitation to initiate appropriate TB therapy leading to delays in starting therapy, use of inadequate regimens, and sub-optimal dosing [8, 9]; and a host of other psychosocial and socioeconomic factors [10].

Pregnant women are vulnerable to all forms of TB, including rifampicin-resistant disease (RR-TB), but there is little documentation regarding optimal management of RR-TB in this high-risk population. Furthermore, few data exist on the lived experiences of pregnant women receiving care for RR-TB. RR-TB is defined as TB disease caused by mutated strains of *Mycobacterium tuberculosis* that render rifampicin ineffective, and in 2018, approximately half a million individuals became sick with this type of TB [11]. RR-TB treatment requires the use of multiple (four to seven) second-line medications for a period of nine to 24 months [12]. Treatment success rates for RR-TB are about 55% globally, and the second-line medications are associated with numerous adverse events—including hearing loss, peripheral neuropathy, and

psychosis [13]. Although there are no estimates or reports of the number of pregnant women who become sick with RR-TB each year, given the age and gender distribution of TB and RR-TB, tens of thousands of women in their child-bearing years are at risk of developing RR-TB annually. Despite this, fewer than 100 pregnant patients have been reported in the literature, many of whom received care prior to 2010 [14–16]. Data from these small cohorts show that RR-TB can be treated effectively during pregnancy with good health outcomes for both the women and their children [17, 18], but a stronger body of evidence on the management of RR-TB and pregnancy is needed [19].

While there is a clear acknowledgement among TB experts that focused work is necessary to better understand how to provide high-quality care to pregnant women living with RR-TB [20], much of the proposed research agenda examines dosing and safety of second-line drugs during pregnancy, including the newer agents bedaquiline and delamanid [21]. In addition to this research, a small body of evidence suggests that pregnant women with RR-TB face a host of additional health and social challenges, including discrimination manifested toward them by health care providers [22]. This discrimination, based on a fear of RR-TB transmission, can result in sub-optimal care and isolation during delivery and the post-partum period and occurs at a time when these women are trying to adapt to two novel identities that are part of their care experiences: that of a "prospective mother" and that of a "RR-TB patient." These identities can contradict one another at times, lead to difficulties negotiating the health care system, and could lead to worse treatment outcomes. This "dual identity" phenomenon has been reported with HIV-positive pregnant women [23] and women diagnosed with breast cancer when pregnant [24]. For example, a woman with breast cancer who is in need of chemotherapy has to weigh up whether or not to receive this treatment, knowing that it might hurt the child (mothering role) but a delay might worsen the cancer (patient role). In both these populations, specific interventions have been necessary to provide optimal support to women, including adapted counseling, integrated care, and specialized training for health care providers.

Understanding the actual care experiences during treatment from the point of view of the person receiving care is crucial to improving services that people receive [25]. This is especially true for diseases where treatment is complicated, including RR-TB [26]. In order to better understand the treatment journeys of pregnant women with RR-TB—and identify areas in which tailored interventions are needed—we conducted a qualitative study among pregnant women living with RR-TB in KwaZulu-Natal, South Africa.

## Materials and methods

### Study design

This was a qualitative study generating data using open-ended interviews among a convenience sample from a population of pregnant women receiving treatment for RR-TB between January 2017 and December 2018.

### Study setting and population

The purpose of the study was to describe the phenomenon of receiving medical care for both pregnancy and RR-TB, and this required working with a population of women who were both pregnant and living with RR-TB. KwaZulu-Natal province has a high burden of both HIV (18.1% in the general population and 44.4% among women in antenatal clinics [27]) and TB (524.4 per 100,000 population [28]). The qualitative study was part of a larger cohort study of pregnant women receiving treatment for RR-TB with the both the traditional second-line drugs and as well as the newer anti-tuberculous agent bedaquiline [29]. This cohort of women

were treated at King Dinizulu Hospital Complex. A convenience sample of 17 women was selected from this larger cohort of 108 women.

## Data collection and analysis

A sample of 17 women participated in open-ended interviews using a semi-structured guide (See S1 Appendix) designed to ask them about their experiences receiving treatment for RR-TB while pregnant and during delivery. All interviews were conducted in the language in which the participant felt most confident (isiZulu or English). Twelve of the interviews were recorded and transcribed into English for analysis. Five of the women did not have their interviews recorded and detailed notes were taken during the interviews instead.

Data analysis used either the interview transcripts or the interview notes and was based in grounded theory which centers the analysis on the accounts of the study participants [30]. The grounded theory approach began with the general question "what is it like to be treated for RR-TB while being pregnant?" and after the first interviews were done, more specific questions and themes emerged, specifically around how the care experiences shaped women's perceptions of themselves, their children, and the health care system. A thematic network analysis was performed to analyze the interview transcripts and notes [31, 32]. The analysis was inductive and iterative in that interviews were transcribed immediately after the interview, transcripts were reviewed by the team, and the interview guide updated to reflect new information. After an initial review of the data during which participants described the ways in which both their identities and activities were differentially shaped by their care experiences for their pregnancy and RR-TB illness, a coding system focused on a "pregnancy identity" and an "RR-TB identity"—as well as their areas of commonality—was developed by one study team member (JF). For this analysis, "identity" was defined as "an evolving, context-sensitive set of self-constructions" derived from an individual's feelings and experiences [33], and in this study the focus was on the experiences of the women during medical care. The analytic framework which emerged from the inductive analysis was verified/modified by another author (ML), and the first 10 interviews were analyzed. Discrepancies were resolved via discussion and there was agreement among all study team members on the final analytic framework used. Interviews were halted after the initial 17 participants since inductive thematic saturation had been reached (determined by two team members, JF and ML) [34], as no new codes or themes were emerging in the dataset [35]. Data collection, analysis, and reporting for this qualitative study followed the consolidated criteria for reporting qualitative research (COREQ) guidelines [36].

## Ethics

Written consent was obtained from all the patients willing to participate in the study. The consent included participation in the interview and digital audio recording, the voluntary terms of involvement in the study and the assurance of confidentiality and anonymity. Patient anonymity was maintained by identifying each patient by a unique identification number. Ethical approval was obtained from the South African Medical Research Council (SAMRC) Ethics Review Committee (EC017-6/2016) and the KwaZulu-Natal Health Research Committee.

## Results

Seventeen women were interviewed. Their mean age was 28 (range 19–38) years and 14 (82%) were HIV-positive. For two of the women interviewed this was their first pregnancy and for the remaining 15 women, this was between their second and fifth pregnancy.

In terms of the "pregnancy identity" and "RR-TB patient identity" that emerged as part of their care experiences, four key themes were identified that appeared to have elements that

were contradictory to one another (contradictory areas): 1) the experience of physical symptoms or changes; 2) the experience of the "mothering" and "patient" roles; 3) the experience of the care their received for their pregnancy and their RR-TB; and 4) the experience of community engagement. There were also three areas that overlapped with both roles and during which each identity was negotiated/reinforced (overlapping areas): 1) faith; 2) socioeconomic issues; and 3) long-term concerns over the child's health (Table 1). Each of these will be described in more detail below. Finally, participants were asked what advice they would give to other women in similar circumstances and their replies are also described.

Before describing these specific experiences, however, it is worth noting that adapting to these two different identifies was difficult, and some of the women reported being in denial

**Table 1. Analytic framework for understanding care experiences for RR-TB and pregnancy.**

| Issues pregnant women with RR-TB must contend with | Identities | Examples (Quotations provided in text) |
|---|---|---|
| **Issues causing conflict between the pregnancy identity and the RR-TB patient identity** | | |
| Antenatal care | Pregnancy: Pregnancy threatened by RR-TB disease and treatment | Possible pregnancy termination or miscarriage, congenital malformations |
| | RR-TB patient: RR-TB care threatened by pregnancy | Worsening of RR-TB disease, risk to successful treatment outcomes |
| Physical symptoms | Pregnancy: Physical symptoms due to pregnancy | Fatigue, shortness of breath, nausea and vomiting could be due to pregnancy; attribution to incorrect cause could lead to delayed diagnosis, sub-optimal treatment |
| | RR-TB patient: Physical symptoms due to RR-TB and/or medication side effects | Fatigue, shortness of breath, nausea and vomiting could be due to TB or side effects of medication; attribution to incorrect cause could lead to delayed diagnosis, sub-optimal treatment |
| Mothering role | Pregnancy: Mothering role complicated by RR-TB disease and treatment | Disease itself poses risk to child, treatment for disease also poses risk to the child |
| | RR-TB patient: Illness role complicated by impending motherhood. | Certain drugs may not be given if they are perceived as being dangerous to the unborn child |
| Community support | Pregnancy: Practice of community support during pregnancy and after childbirth threatened by RR-TB disease | Usual support networks may avoid those with RR-TB due to stigma, fear of contagion. |
| | RR-TB patient: Practice of social distancing and isolation reduces potential community support. | People with RR-TB may avoid others due to fear of contagion or worries they will be treated badly |
| **Issues common to both the pregnancy identity and the RR-TB patient identity** | | |
| Socio-economic challenges | Both antenatal and RR-TB care threatened by socio-economic challenges | Pregnant women with RR-TB have to access health facilities far more often as they seek both antenatal and RR-TB care. A loss of income with increased expenses exacerbates their economic vulnerability. Increased expenses include paying for transport to access a health facility and childcare. |
| Long term concerns for child's health | The women worried about the effect of the RR-TB drugs on their infant's growth and development. They also worried that they might infect their infant with TB. | Worries that the child will develop TB later in life or that the effects of the medications will appear later in the life of the child |
| Faith | Most of the women reported a dependence on "God" or the ancestors at this time | Assuming the outcome of the treatment and pregnancy are "in the hands of God", relying on prayer |

about their RR-TB diagnosis—especially if they had minimal TB symptoms. So complicated was the idea of taking on both the patient and the pregnancy identities that some women who were already on RR-TB treatment or who were earlier in their pregnancies did not wish to continue being pregnant—that is the care experience was so complicated they thought about eliminating one of their identities. The complexity of this dilemma was, at times, exacerbated by health care providers. In the following example, a participant was given conflicting advice from clinical staff regarding termination of pregnancy (TOP). Firstly, at the antenatal clinic, the providers strongly recommended she consider TOP:

> *"They said things like, 'this baby will be disabled, [the]father of this baby will leave you once he sees that the baby is disabled. The community you live in will look at you with a disabled baby. Would you like to be born with a disability after being warned about being disabled during pregnancy?'*

(Participant 2)

Later, on being admitted to the tertiary referral hospital, she was reprimanded for considering such an option. Ultimately, with the support of the RR-TB hospital staff she opted to continue her pregnancy and her RR-TB treatment. A second participant who considered TOP was told by her RR-TB physicians they could not assist her, despite national legislation to the contrary.

## Contradictory areas that were part of the care experience

**1. Experience of physical symptoms.** Most women interviewed were pregnant or sought care for their pregnancy prior to being diagnosed with RR-TB. Some were experiencing symptoms that could have been caused by TB, but which they initially attributed to their pregnancy. As one participant noted:

> *"Every morning when I must wake up to prepare to go to work, I would have that problem [fatigue]. But I thought it was because of the pregnancy, but when time went, I noticed that I was losing weight, lost appetite."*

(Participant 9)

Attributing their TB symptoms to pregnancy may have led to a delay in seeking care for RR-TB. The participants did report, however, that it was most often the nurses providing antenatal care (ANC) who suspected that they might have TB and requested that they have their sputum tested. As one participant noted:

> *"I got to the clinic and told them that I was there to make a maternity card. They then sent me there to test for HIV and I went. I was also asked for sputum, I did."*

(Participant 10)

Of note, some participants reported that while they were being treated for RR-TB and developed adverse events—especially nausea and vomiting—these were attributed to their pregnancy and may not have been assessed or managed in the same way as they would have been in non-pregnant patients.

**2. Experience of "mothering" and "patient" roles.** Taking on the role of the patient and the mother at the same time was difficult. Most of the time, having to embrace both these

identities at the same time detracted from what might have been experienced had these identities been taken on separately.

For many women in this study, the mothering role was diminished by the patient role in several ways. First, many of the women were already mothers, and when they had to be hospitalized for RR-TB treatment—usually because they were pregnant—they felt they were unable to be the mothers they wanted to be for their existing children. As one woman noted:

*"I was told I was going to be admitted! Then I said, 'it cannot be. My children are at school and I am the older person that is supposed to come back home, and the house keys are with me'. But they said I will be admitted. . .My heart was very sore. It was not nice at all."*

(Participant 6)

Other women reported that they felt their roles as mothers were diminished because the medicines they were taking or the RR-TB itself might harm the unborn child. As one woman noted:

*"I asked myself a question, what will happen to my baby, you see. Because I am taking this treatment. Maybe, he will be born not well, or he will be fine, I don't know. I had many questions."*

(Participant 16)

Some of these women reported that their fears were allayed when the pregnancy progressed as their previous pregnancies had. However, in others whose pregnancy felt different—that is they had more symptoms or felt the child was moving less—their concerns that there would be something wrong with the baby increased. This anxiety was exacerbated in those who saw other patients miscarry or die whilst they were hospitalized.

Some women also reported that adapting to the patient role included adapting to that of an inpatient, as according to national policy, RR-TB treatment initiation whilst pregnant required hospitalization. This was a more complicated patient role than if they had been able to be treated outside of the hospital and made it more difficult for them to receive RR-TB treatment.

*"I cried because. . .. I had told myself it will be easy for me to take the treatment and take them and go home. So, when they found I was pregnant I was told that I'd have to be admitted."*

(Participant 10)

*"Then she said, it is a must that you go to hospital there is nothing you can do, because you are pregnant you cannot take treatment outside the hospital, you must take it in hospital until you give birth then you will be assessed after giving birth."*

(Participant 17)

However, some women reported that the pregnancy was the reason they felt that their RR-TB had been discovered, and in this way, the pregnancy was an integral part of the patient identity:

*"I was stressed about being pregnant. . .but it helped me because, I would not have known that I have MDR, I had no signs of having MDR, I was not coughing I was well. But I always say; this child came to reveal this, this disease was discovered."*

(Participant 11)

Many participants reported that both their mothering and their patient roles were enhanced when they viewed taking treatment as a way to protect their unborn children. That is, they were more likely to take treatment (enhanced patient role) because they felt it made them better mothers since it enabled them to protect their babies from RR-TB disease.

*"It was very painful, I had to think about the baby, [the medicine] was strong, all the side effects but I was taking it. It was a lot to take because I had to take double for her."*

(Participant 13)

*"But then again I remember what will happen to my children if I die, I struggled when I was in hospital how much more will they suffer if I die. If I fail to live for them."*

(Participant 12)

For some women it was a challenge balancing the potential risk of the RR-TB medication on the developing fetus with the need to take treatment to save her life and the life of the unborn child. Complicating this was a dearth of information on the risks of most medications used in treating RR-TB. The complexity of some of this information is difficult to comprehend for a pregnant woman, struggling to process a diagnosis of RR-TB.

**3. Perceptions of quality of care.**   Most of the women reported that they felt their care during pregnancy was sub-optimal because of their RR-TB. They also reported aspects of their RR-TB care that were sub-standard compared with national and international recommendations. Although these women did not necessarily know their care was not following national norms, they did report noting that aspects of their RR-TB care were different to that others were receiving and assumed this was due to their pregnancy. Some women felt that after their RR-TB diagnosis, the antenatal care (ANC) providers did not want to see them:

*"At the clinics they don't even want to spend time with us. She didn't want to talk to me. . .. The way she spoke to me was not how a nurse should speak."*

(Participant 12)

However, most of the perceived problems with pregnancy care occurred around the time of delivery, where several women reported being left alone without being checked on or examined properly. They reported not being provided with food for themselves or for their babies, and it was only when their families complained that they were attended to by nurses. Another woman reported that the physician delivering her baby became abusive when he found out she had RR-TB:

*"I was treated well until they discovered that I have [RR-TB]. . .they started having a problem once they look at my card. . .the doctor that was helping me give birth, he had anger, he was worse with me because he had learned that I had MDR. I do not wish for anybody to experience what I went through at nursery."*

(Participant 12)

In fact, most of the women reported being discriminated against by their perinatal care providers. This discrimination was manifest through refusal to provide care, fear-based infection control practices (such as donning gloves to speak with the women and putting them in unnecessary isolation), and stigmatizing actions and language, often when referring to the unborn child. As one participant reported:

> *"My results came back. When they came back, I noticed the way the nursing sisters acted when they told me I had MDR and they began to wear gloves, I didn't even know what MDR was then."*

(Participant 6)

Another reported that the nurses "yelled" at her for becoming pregnant:

> *"Okay I admit I made a mistake of becoming pregnant, but why she has to treat me like that?"*

(Participant 9)

Another reported that the ANC nurses kept referring to her child as "that thing" (Participant 2). One woman reported that she overheard the doctor speaking about her during her delivery:

> *"He said; if the mother is not dying today, the baby will. He said; between the two of them one of them will not make it today. And when I heard that he was speaking to another nurse."*

(Participant 12)

Four participants (#s11, #2, #3, #4) described being isolated at the time of or just after delivery by health care workers fearful of RR-TB. They described being kept alone in an 'isolation room' where no health care workers or cleaning staff entered. Their food was put on the floor outside their room and visiting family members had to remove garbage such as dirty nappies.

One participant had a different experience and felt she was treated well and kindly by the ANC staff during her delivery. She reported that they were upset that they did not know she was an RR-TB patient when she came to deliver her baby, but that their frustration was with the other providers and not with her (Participant 11).

In terms of RR-TB care, most women described problems with the routine TB services at local health clinics where they were initially diagnosed (although the women themselves may not have been aware of this). Such practices included being started on treatment for drug-susceptible TB even though the laboratory results showed they had RR-TB and being given incomplete RR-TB treatment regimens in order to avoid medications that might have an adverse effect on the developing fetus. However, the most common RR-TB treatment malpractice was to delay treatment initiation (at times for several weeks) until admission to a tertiary referral hospital for inpatient treatment. As one participant noted:

> *"They said they called [the local hospital], they said the doctor there he cannot take me. They said because I was pregnant as well. They then said, they called here, [the tertiary referral hospital]. It was December, the doctor said he was not doing admissions during December, I must come in January. But they gave me treatment, Rifafour [treatment for susceptible TB] to use in the meanwhile."*

(Participant 14)

Of note, however, most women in this study reported that once they arrived at the tertiary referral hospital, they felt well treated and got excellent care:

*"No, coming here [the tertiary referral hospital] we were warmly welcomed at the reception. They took us to the wards and showed us to the doctors, the nurses accepted us."*

(Participant 10).

Women also reported receiving better information about RR-TB once they were admitted to the tertiary referral hospital compared to that which they received at their local clinics. Most of this information, however, was focused on the side effects of the RR-TB treatment regimens and the possible impact of the medications on their unborn children.

*"I got better explanation here at [tertiary referral hospital], they didn't explain anything at the clinic."*

(Participant 12)

Most of the women who participated in this study were receiving the newer TB drug bedaquiline and reported being concerned about the lack of information on the use of this newer drug in pregnancy:

*"He told me that he had never given bedaquiline to a pregnant woman before, but he had given it to a monkey! So, I then asked the doctor if he is doing research through me?"*

(Participant 10)

It is noteworthy that some of the study participants reported experiencing discrimination from RR-TB providers because they were pregnant. This was largely experienced as negative attitudes expressed by staff toward the women for becoming pregnant.

*". . . . . . . .every time when I have to come to the clinic I get stressed out, especially if I have to be seen by the doctor who doesn't know that I am pregnant. Eish I get very stressed out because I [am] frightened that s/he is going to shout at me."*

(Participant 9)

Although some women reported that their fears of being judged by the RR-TB providers were so significant that they dropped out of RR-TB care, in general, there was less discrimination reported from providers at the tertiary referral hospital, with some women reporting the physicians and nurses there as being among their main sources of moral support.

*"So, the doctor supported me, he stood by me, he was my pillar because I was always crying. I'm not sure if it was because I was pregnant."*

(Participant 10)

*"[The nurse at the referral hospital] is the person who spoke sense. I used to cry daily, I didn't go through the day without crying, I cried for 2 weeks. I slept, woke up to bath and go back to bed to cry. She gave me counselling the way she could, and my heart started healing."*

(Participant 12)

**4. Experience of community engagement/isolation.** Pregnancy is usually a time of increased engagement with the larger family and social communities and for many women in our cohort, this was the case. However, others felt that the community engagement they

expected was not provided because of their RR-TB disease. This was largely due to their hospitalization at the tertiary referral hospital and being told to stay away from people as they might infect others. As one participant noted:

> "...when people see that you are sick, they change and turn their backs on you and not treat you like before."

(Participant 9)

Some of the women reported isolating themselves from people outside of their immediate families:

> "Friends, I do not want to lie, I would not know because when I came back from the hospital I was staying indoors."

(Participant 12)

Most of the women reported that their mothers, together with the father of their unborn child were a source of social support that enabled them to navigate their two identities. The most common source of social support (sometimes unexpected) was the father of the unborn child. In our setting, patriarchy is very common, and the care of children and the sick considered the responsibility of women.

> "The father of my kids is very supportive. As I said, It's just me and my mother, the father of my kids is very supportive and he is the one who was supporting me whilst I was in hospital and even now I am still with him, he did not change or treat me differently. His focus was on me to get better and take care of the kids. He is the person who made sure that they are okay."

(Participant 10)

Several participants reported that their colleagues and co-workers provided them with social support:

> "My colleagues did not do that [avoid me], instead they supported me, checking up on me even when I was still here [in hospital]."

(Participant 6)

### Overlapping areas that arose from the care experience

**1. Faith.**   In the absence of adequate information about their RR-TB disease, to help cope with the challenges of taking on the dual identities, most of the women reported a dependence on "God" or the ancestors at this time. This was a positive "overlapping" issue and was reported by most women in this study:

> "God can surprise you with anything, because surprises are happening here in the universe. It's something that you cannot block because even if you are not on MDR-TB treatment [and] if God had planned to take the baby away from you, He was going to take it even if you are not on MDR-TB treatment."

(Participant 9)

*"What [the nurse] told me was; if a mother is a parent, that mother does not cry. There is one way that a mother cries, it is through prayer only."*

(Participant 12)

Not all women in the study relied on "faith" and two women reported substance and alcohol use both before and during their pregnancies as a coping mechanism.

**2. Socioeconomic issues.** Almost all the women reported that socioeconomic issues and challenges were part of both of their identities as pregnant women and as people living with RR-TB. This was a negative cross-cutting issue that added additional challenges to their treatment journeys. As one woman reported:

*"You see there were a lot of things. How will the baby grow? Financial issues, everything, everything was a mess my sister. That's it."*

(Participant 15)

Often, socioeconomic issues were exacerbated by the forced admission to hospital for initiation of RR-TB treatment:

*"I was fired due to being admitted in hospital. . . they explained that there is nothing they can help me with as I am in hospital. They have to replace me with someone else because the work needs to continue regardless."*

(Participant 10)

Commonly reported challenges included the cost of transport to health facilities, loss of regular income during RR-TB treatment, and increased household expenditures due to pregnancy and the birth of the child. Some of the women reported that receiving socioeconomic support from the health care system alleviated many of their needs and concerns and that they appreciated this aspect of care. As one participant reported:

*"They were caring, they gave me food, cash to manage me. The children, they checked. There was one nurse that was too lovely. She screened them. She gave me extra money and the other counsellor, he also gave me."*

(Participant 13)

**3. Long-term concerns over child's health.** Most of the women continued to have concern about the long-term health of their children. As one participant stated:

*"You see, sister, there is nothing hard like being alive. You know when this baby is sleeping, sometimes I look at her and think maybe she is dead. . .. I often place my hand on her nose to feel if she still breathes. I am afraid for her. I still have that feeling that she will not get to 5 years, or 4 years. . . I think the TB is still hiding, it has not yet been discovered. Maybe as time goes by, I will be told that she has died."*

(Participant 12)

Of note, these concerns were alleviated when participants saw their babies developing normally and meeting their growth and development milestones:

*"Maybe she is weak, because that is what I was always telling myself; that she won't be strong because I was pregnant with her while taking treatment. That's what I was telling myself. But that did not happen, she was not weak. I was satisfied with everything she was doing."*

(Participant 12).

Seeing the children do well also helped enhance the mothering role for many of the women in the study. As one participant reported:

*"She is beautiful. I always tap myself about giving birth to such a beautiful baby."*

(Participant 14)

## Advice for others based on the care experience

Finally, the participants in this study were all asked what advice they would give to other pregnant women with RR-TB as well as what they would tell health providers about providing care to such women. The main advice offered focused on the need for resilience to navigate the care system and the often conflicting identifies that resulted from these care experiences.

*"Take the treatment and be brave. . . because the important thing is to be willing to go through it and tell yourself that you will survive, this is your life and don't look at what others will say and whether or not they will judge you for being on treatment. I would tell her that she is not doing it for anyone else but herself."*

(Participant 10).

And another participant noted:

*"I would try to advise her, as a person who have been through that situation. I would explain to her that she must take the pills, she must take care of herself and do what she is told to do."*

(Participant 9)

In terms of how to improve the experience of pregnant women with RR-TB, most women in the study reported that care could be improved simply by treating them as "normal" or "ordinary" people:

*"She must be treated like anyone else. Because she has MDR people must not say; this one has MDR, because they do not know how she got it. We travel by taxis, breathing from the same air, you don't know whether you have or not. Yes. They should treat you like anyone."*

(Participant 11)

Another participant recommended the care be taken one step further:

*"They have to be treated with caring, love. They have to be clean. They have to be treated the right way because MDR is a very bad sickness. It's a killing disease. They will have to be treated with extra care."*

(Participant 13)

## Discussion

Women who participated in this study reported several challenges resulting from their care experiences as pregnant women with RR-TB: overall, these challenges increased their sense of vulnerability. Some of these challenges arose from their experiences of receiving care in contradictory roles as women with both new life and potential deadly bacteria growing inside them. These problems were exacerbated by a health care system that was not comfortable or supportive in helping women negotiate these two roles. From the physical symptoms they experienced to the advice they received about optimal treatment, women were often unsure about what exactly was happening to them and what they should do about it. Their continued engagement in care required resiliency, often expressed as "bravery". At times health care providers were ill-informed: often they were frightened or angry and treated the women in a discriminatory fashion. Left to negotiate this confusing time period, women turned to faith, their mothers, and the fathers of their unborn children. While they showed resilience in the face of remarkable health and social challenges—and significant socioeconomic burdens as well—the results of this study show multiple ways in which their care could be improved (or, as many requested, simply be the same as that of "ordinary people").

The dual identities developed by these women in response to their care experiences are not unique to RR-TB and pregnancy. Similar challenges have been reported among women living with HIV and cancer during pregnancy, where decisions have to be made about what is best for the health of the mother and the heath of the unborn child, often in the face of limited data to guide that decision making [37]. With HIV, however, there appears to be a much greater comfort on the part of women and their providers in holding these two identities together in the same space, perhaps because many early HIV therapeutic interventions were targeted at pregnant women [38]. It is notable that many women in our study were also dealing with HIV but did not report problematic interactions around their HIV treatment. Lessons learned from the HIV pandemic about harm reduction, integrated care, respectful engagement, and community building could be adapted to the field of RR-TB, offering women a more dignified experience at this transitional time in their lives [39].

In addition to integrated care—which studies show is still a long way from being available to pregnant women with infectious diseases [40]—the results of this study show that besides additional psychosocial and socioeconomic support pregnant women with RR-TB would benefit from tailored counseling aimed at addressing their specific needs in both the ante- and post-partum periods. Such counseling, could include how health care providers best communicate with pregnant women, understanding their rights around delivery, and how to manage common adverse events. Women could be offered participation in support groups after delivery where they and their children could meet with other mother/child pairs, both for social interactions and to share their ongoing concerns about child development. Given that fathers of the babies were mentioned as a strong—but unanticipated—source of support, they should also be more formally engaged in women's treatment journeys. Socioeconomic support is key during the antenatal and peripartum periods since many women have increased expenses—due both to illness and to their new babies—at a time when they are unable to work outside their homes. Table 2 summarizes features that could be included in an optimized care package for pregnant women, based on the results of this study.

There is an urgent need to tackle the discrimination participants reported. Health care provider education is necessary—and it is notable that most women felt more comfortable at the tertiary hospital that had experience dealing with pregnant women who had RR-TB. Such education is unlikely, however, to be sufficient to enact change, and formal channels need to be established for women to safely report discrimination and for it to be remediated. The impact

**Table 2. Recommended elements of an optimized care package for pregnant women with RR-TB.**

| Area of Concern | Optimizing Services |
|---|---|
| Antenatal care | Integrated care for pregnancy and RR-TB provided by specialized providers with expertise in both areas. |
| Physical symptoms | Early identification and assessment of symptoms by trained health care providers; |
| | Diagnostic assessment provided free of charge. |
| Mothering role | Specialized counselling and support; |
| | Engagement of peer support networks of other women who have been treated for RR-TB during pregnancy. |
| Community support | Capitalize on the role of the father of the baby together with the mother of the pregnant woman; |
| | General community education and sensitization about TB; |
| | Engagement of peer support networks of other women who have been treated for RR-TB during pregnancy. |
| Socioeconomic challenges | Provision of basic package of support services, including transport, nutritional support, and provision of essentials for the new baby. |
| Log-term concerns for child's health | Provision of free, long-term medical follow up for children born to women during treatment for RR-TB. |
| Faith | Engagement of churches and faith-based organizations in the care of women who are pregnant with RR-TB; Use of prayer as part of counselling. |

of discriminatory isolation should be addressed immediately given the physical dangers it poses to peri-partum women. It may also have a significant psychological impact on women, many of whom come from community households with large families.

This study has several limitations. It is a qualitative study done among a small group of women and offers a rich description of their different experiences of having RR-TB during pregnancy, but it was not designed to achieve representivity. It utilized a convenience sample and thus may not have captured a diverse range of experiences. Some of the interviews were not recorded and this may have led to missing important quotations or themes that might have been uncovered in word-for-word transcripts. Although the interviews were open-ended, they did focus on the experiences of RR-TB treatment during pregnancy and may have missed other crucial life experiences during this transitional time period in participants lives. Finally, and as part of the tradition of reflexivity that is essential in doing qualitative research, we note that two of us are engaged in providing care to people with RR-TB as medical providers and this may have impacted our understanding, analysis, and description of the experiences of the women who participated in this study.

## Conclusion

Despite the limitations, the study has important findings that should change the current approach to the treatment of RR-TB among pregnant women. There is an urgent need to include such women in ongoing clinical and operational research studies as well as to develop pregnancy registers, both so they can benefit from scientific progress but also so information can be collected about optimal therapy—including the efficacy, safety, and dosing of second-line TB drugs. However, there is an equally pressing need to develop optimized packages of support to enable more positive care experiences for pregnant women with RR-TB. Such pack-ages should include integrated care provided by trained medical professionals with skills in managing both RR-TB and pregnancy; utilization of peer support networks; optimization of social support, including the fathers of the children and the mothers of the pregnant women; provision of socioeconomic support; long-term medical care for children born to women who

were pregnant during treatment for RR-TB; and broader inclusion of faith-based organizations and practices during treatment. It is time for the TB community to become both compassionate and bold in addressing all the needs of this vulnerable group.

## Supporting information

**S1 Appendix.**
(DOCX)

## Author Contributions

**Conceptualization:** Marian Loveday, Jennifer Furin.

**Data curation:** Sindisiwe Hlangu.

**Formal analysis:** Marian Loveday, Jennifer Furin.

**Funding acquisition:** Marian Loveday.

**Investigation:** Sindisiwe Hlangu.

**Methodology:** Marian Loveday, Jennifer Furin.

**Project administration:** Marian Loveday, Sindisiwe Hlangu.

**Resources:** Marian Loveday.

**Supervision:** Marian Loveday.

**Validation:** Marian Loveday, Sindisiwe Hlangu.

**Writing – original draft:** Jennifer Furin.

**Writing – review & editing:** Marian Loveday, Sindisiwe Hlangu, Jennifer Furin.

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
