## [Decision Letter · Decision Letter 0]

7 Aug 2020

PONE-D-20-13335

"Take the treatment and be brave”: A qualitative study of the experiences of pregnant women with rifampicin-resistant tuberculosis

PLOS ONE

Dear Dr. Loveday,

Thank you for submitting your manuscript to PLOS ONE. After careful consideration, we strongly feel that it has merit but does not fully meet PLOS ONE’s publication criteria as it currently stands. Therefore, we invite you to submit a revised version of the manuscript that addresses the points raised during the review process.

We look forward to receiving your revised manuscript.

Kind regards,

Jennifer Zelnick

Academic Editor

PLOS ONE

Journal Requirements:

Additional Editor Comments (if provided):

Thanks for this submission on a topic where there are too little published data. The reviewers have provided detailed observations. In particular, editing the abstract to better reflect the themes and including recommendations are strongly suggested to improve this paper.

Reviewers' comments:

Reviewer's Responses to Questions

**Comments to the Author**

1. Is the manuscript technically sound, and do the data support the conclusions?

Reviewer #1: Partly

Reviewer #2: Yes

Reviewer #3: Yes

2. Has the statistical analysis been performed appropriately and rigorously? 

Reviewer #1: N/A

Reviewer #2: N/A

Reviewer #3: N/A

3. Have the authors made all data underlying the findings in their manuscript fully available?

Reviewer #1: No

Reviewer #2: Yes

Reviewer #3: Yes

4. Is the manuscript presented in an intelligible fashion and written in standard English?

Reviewer #1: Yes

Reviewer #2: Yes

Reviewer #3: Yes

5. Review Comments to the Author

Reviewer #1: Thank you for asking me to review the paper entitled: “Take the treatment and be brave”: A qualitative study on the experiences of pregnant women with rifampicin-resistant tuberculosis. (PONE-D-20-13335)

I truly enjoyed reading the paper, and my comments below aim to strengthen its presentation.

Summary of research and overall impression

The paper is based on qualitative data generated through semi-structured interviews with 17 women with rifampicin-resistant tuberculosis (RR-TB) who experienced pregnancy concurrently to being treated for RR-TB. The study was nested within a cohort study of pregnant women, which appears to have compared exposure outcomes between traditional second-line drugs for RR-TB and the newer anti-tuberculous agent bedaquiline. The purpose of the qualitative data generation is stated as wanting to better understand the different self-perceived roles and treatment journeys of pregnant women with RR-TB, in order to identify where tailored interventions are needed.

This study has relevance within the context of ever-increasing commitments to transforming health systems to being patient-centred. The study contributes to ‘giving voice’ to the patient experience of care. The main utility of the study lies in documenting the care experiences from the perspective of pregnant women with RR-TB, and the challenge it poses for the responsiveness of the health system.

The article presents as main findings: (1) the dual identity of women who experienced pregnancy concurrently to being treated for RR-TB; (2) various care experiences, mostly negative as a result of the RR-TB diagnosis, and some positive; and (3) coping strategies that the women used.

I do believe that given how the current version of the article reads, the emphasis of the paper should be on ‘care experiences’ and the issue of dual identities should be addressed in relation to the care experiences. The rationale for this recommendation: exploring experiences to reveal the meaning they have for identity lends itself to a phenomenological study design. Given the emphasis in the paper on the management of RR-TB in pregnancy and the need to tailor care interventions for the specific needs of pregnant women with RR-TB, lends itself more to the grounded theory approach the authors stated they employed. Coping strategies would then need to be discussed in the light of person-centred models of care.

Specific comments/Recommendations

Given the qualitative approach to the study, I have used the COREQ checklist for reporting qualitative research (Tong, Sainsbury and Craig, 2007) in appraising this paper and in compiling this review.

1. Title: consider deleting ‘a qualitative study on’; consider inserting ‘care’; consider adding a context - to read: “Take the treatment and be brave”: Care experiences of pregnant women with rifampicin-resistant tuberculosis in Durban, South Africa.

2. Key words: consider adding ‘qualitative study’ or ‘qualitative research’.

3. Abstract:

a. The background in the abstract is ambiguous. [“Little data on the management of pregnant women with RR-TB” could be understood to mean the clinical management of RR-TB. “… or their experiences” could mean the effects/side-effects of the clinical management. The background does not foreground the research problem that this paper seeks to address – the care experiences of women who are both pregnant and on treatment for RR-TB.

b. Methods – the purpose of the COREQ guidelines is to check the quality of reporting of studies employing qualitative methodologies, and do not guide purposive sampling. The qualitative study design/methodological orientation employed in the study should be stated in the methods. The word ‘documents’ is misleading, as the primary and only source of data was semi-structured interviews, which were then either transcribed (changed from oral data to textual data) or annotated. Iterativity usually refers to the cyclical nature of data collection and data analysis, and usually is employed to enrich data collection so that saturation of meaning/information power is achieved (see Hennink et al. 2016; Malterud et al 2015). In implementing a thematic network analysis, authors should state whether they adopted an inductive, deductive or hybrid approach in identifying the basic, organising and global themes.

c. Results: The presentation of results should align with the thematic network analysis adopted. Is “take the treatment and be brave’ the global theme? How do the organising themes illuminate the need to ‘be brave’? What does being brave ultimately mean for the care experience (note the singular versus the plural use) of pregnant women with RR-TB?

d. Conclusion: reads very generically and does not particularly align with the purpose and results of the study.

4. Introduction: has a strong emphasis on the clinical aspects of TB and RR-TB in pregnancy and does not adequately bring under focus/foreground the need to explore the patient perspective/voice/experience of care. Further, the introduction may well pre-empt the findings in highlighting the dual identities. Unless the authors indeed acknowledge the dual identities upfront, and then focus the study on characterising the care experience given the dual identities.

5. Study design – the authors do not identify the methodological orientation underpinning the study. They do allude to a grounded theory approach to data analysis, but do not provide convincing evidence that they were indeed faithful to the methods of a grounded theory study design.

6. Study setting and population –

a. The authors identify the population that was studied as pregnant women receiving treatment for RR-TB. Usually qualitative designs are employed to study phenomena rather than populations. Phenomena are studied from the perspective of a particular group or groups.

b. There is insufficient detail regarding the sampling: what purposive sampling strategy was employed? What was the rationale for the sampling strategy employed? What were the sampling criteria used to select the 17 pregnant women from the 108? Were all women pregnant at the time of recruitment? How were the respondents recruited?

7. Data collection and analysis

a. What approach to data collection was used – general interview guide approach or was a standardised open-ended interview conducted? The interview schedule design and the ordering and framing of the questions suggest the latter approach, but in the analysis section the authors state that the interview guide was updated after each round of data collection. What updates were made?

8. Results

a. The results skim over the ‘dual identities’ and quickly turn to the experiences of care – being given advice re: TOP.

b. I am not convinced that each of the themes relate to the phenomenon of dual identities. In my opinion most of what is reported relates to the care experience and speaks very loudly to the nature of care systems required – from compassionate health workers, to intersectoral support systems to care for their social needs, to health information, to integrated care systems, etc. I would recommend that the results be organise to characterise (in organising themes) the care experience.

c. I would recommend that the coping mechanisms identified be discussed in the light of how they can be harnessed to improve on the care for pregnant women with RR-TB, particularly within person-centred models of care.

d. The quotation – take the treatment and be brave – seems to be an encouragement, at minimum to be adherent to care processes, and at best to be full participants in the care process. It may well also speak to the need for peer mentoring/buddy system in the care process, similar to the Mothers to Mothers-to-Be (M2M2B) programme.

9. In the discussion I would opt to focus on the characterisation of the care experience and to even provide a visual diagram of the actions and interactions that influence the global characterisation of the care experience for pregnant women with RR-TB. The visual provided at present is labelled “Analytic framework” so it is not clear if this provided the framework for a deductive analysis or if this was the outcome of an inductive analysis. Generally grounded theory is used to generate an hypothesis, and based on the results, the hypothesis generated would need to identify the aspects that a care package for pregnant with RR-TB would need to address.

10. Conclusion: does not seem to align with the purpose and nature of the study.

11. Limitations: I do not accept the limitation that the study cannot be generalizable to other populations. Firstly, you did not study a population – you studied a phenomenon (poorly defined but nevertheless can be named as ‘care experience of pregnant woman with RR-TB); secondly you implemented a qualitative study to study the phenomenon – qualitative studies do not aim to achieve generalizability – as you correctly stated you sampled information rich respondents to help to saturate the phenomenon. You did not sample to achieve representivity so that you can generalise to a population. In qualitative studies you aim for transferability – and you need to provide a rich context description so that your reader can assess from your description of the context how similar or dissimilar it is to their own, and whether the findings you report may be transferrable to their context. The real limitation is that you do not describe the context of care beyond the medical management of RR-TB. How does RR-TB get diagnosed in pregnant women, where are they assessed first, where are they referred to, what period of hospitalisation is required, what type of personnel cares for pregnant women with RR-TB at each level of care. This is all important background information that contextualises the care experience of pregnant women with RR-TB.

12. As a final note: reflexivity is an important process in studies within the interpretivist paradigm and that generate qualitative data – how was this requirement met in this study?

Reviewer #2: Thank you for such an important paper.

The manuscript highlights the experiences of pregnant women with RRTB. Overall, the pregnancy identity comes across a lot more than the patient identity. although the illness identity is discussed, the extent to which it is complicated by pregnancy does not come across strongly.

This was a study on pregnant women, the authors present the fears that the participants had regarding the health and development of their children (Line 508-509). It is not clear if the participants were re-interviewed after giving birth? or at any other point seeing that data collection was over a 5 year period. Perhaps clarify this under data collection.

The authors point to the challenges faced by regnant women at the ante-natal clinics, perhaps they could also make a suggestion that support be offered at this point and not only after delivery?

The authors also suggest (Line 587) counseling on how best to communicate with health care providers. This suggests a deficiency in the patients. The authors could rephrase this, perhaps to indicate the need for health workers to communicate better with patient, or for patients to assert themselves more, whatever the authors intend to put across.

Thank you

Reviewer #3: This is a qualitative study of pregnant women who are receiving treatment for drug-resistant tuberculosis. Given the paucity of data on this particular group (all published data to date has only 100 pregnant women with DR-TB in total) and the use of qualitative methods, it is novel and worthy of sharing. The authors follow COREQ guidelines, and this is appreciated. I have some suggestions to the authors that I think , if addressed, will improve the manuscript. They are as follows:

MAJOR

(1) Abstract: The manuscript body nicely divides experiences as a pregnant woman and as a patient with TB into those that are contradictory (those in which a woman's role or responsibilities as mother and as MDR-TB patient contradict each other) and those that are overlapping (where actions taken to be the best soon-to-be mother and the best patient one are aligned). The abstract, however, fails to give examples of each of these, and one is left with generalities and statements that are somewhat self-evident. The conclusions fail to provide any concrete or specific ways we can help these women. I'd suggest a rewrite of the Results and Conclusions.

(2) Methods: Please describe how exactly participants were 'purposively selected'? Based on what criteria? To achieve what objectives?

(3) Results: Table 1 is not helpful. It merely lists the age, HIV status, and # of births (?including the present one?) of each participant. These statistics can be summarized in the main text.

(4) Results: Consider replacing the Figure with a Table that summarizes contradictory areas and overlapping areas, perhaps with some brief examples. This will be helpful to the reader , to help him/her organize and capture main ideas. I overall like the visual impact of this type of figure as an analytical framework, but here it doesn't work too well because the words in the left circle and the ones in the right circle are essentially the same, just reversed.

(5) Discussion: This would be enhanced if the authors would provide a prioritized list of ways care could be improved for pregnant women with DR-TB (lines 567-568 could be linked to a table that outlines recommended interventions)

MINOR

(1) The word 'data' is plural, so please make relevant adjustments (e.g. there are few data rather than there is little data)

(2) Lines 120-122- what type of discrimination, and what are the reasons for this discrimination?

(3) Lines 124-125- give a short example of how identities of prospective mother and patient with HIV or cancer are at odds so the reader knows what you mean here. You explain it later in the Discussion, but it would be better to introduce this 'contradictory roles' idea early in the paper.

(4) Lines 133-124- it took 6 years to enroll this 17-person study. Why?

(5) Avoid inflammatory or hyperbolic language (e.g. pressured instead of harassed, vulnerable instead of incredibly vulnerable...); the point is made without that.

(6) Consider putting findings related to termination of pregnancy and also about mandatory hospitalization into the abstract so they get more visibility. These are really important issues and are concrete and illustrative examples of challenges these women face when having the dual role of pregnant woman and patient with TB

(7) How did these 17 women do?

(8) Did the research team organize support groups for these women ? It seems that such a group, with an advocate participating , would be valuable.

(9) Lines 397-398: would assume that women were not always in control of their reproductive health

(10) The authors make a point of stating that it was unexpected that the women's husbands/ partners would play a role in supporting them during this difficult time. Consider describing why this was unexpected.

6. PLOS authors have the option to publish the peer review history of their article (what does this mean?). If published, this will include your full peer review and any attached files.

Reviewer #1: **Yes: **Dr Anna Silvia Voce

Discipline Public Health Medicine

College of Health Sciences

University of KwaZulu-Natal

Durban, South Africa

Reviewer #2: **Yes: **Boitumelo Seepamore

Reviewer #3: No

---

## [Author Response · Author response to Decision Letter 0]

28 Aug 2020

Additional Editor Comments (if provided):

Thanks for this submission on a topic where there are too little published data. The reviewers have provided detailed observations. In particular, editing the abstract to better reflect the themes and including recommendations are strongly suggested to improve this paper.

Response: Thank you for this comment. As you will see in the revised version of the manuscript, we have revised the abstract to better reflect the themes emerging from our study and the recommendations.

Reviewer #1: Thank you for asking me to review the paper entitled: “Take the treatment and be brave”: A qualitative study on the experiences of pregnant women with rifampicin-resistant tuberculosis. (PONE-D-20-13335)

I truly enjoyed reading the paper, and my comments below aim to strengthen its presentation.

Summary of research and overall impression

The paper is based on qualitative data generated through semi-structured interviews with 17 women with rifampicin-resistant tuberculosis (RR-TB) who experienced pregnancy concurrently to being treated for RR-TB. The study was nested within a cohort study of pregnant women, which appears to have compared exposure outcomes between traditional second-line drugs for RR-TB and the newer anti-tuberculous agent bedaquiline. The purpose of the qualitative data generation is stated as wanting to better understand the different self-perceived roles and treatment journeys of pregnant women with RR-TB, in order to identify where tailored interventions are needed.

This study has relevance within the context of ever-increasing commitments to transforming health systems to being patient-centred. The study contributes to ‘giving voice’ to the patient experience of care. The main utility of the study lies in documenting the care experiences from the perspective of pregnant women with RR-TB, and the challenge it poses for the responsiveness of the health system.

The article presents as main findings: (1) the dual identity of women who experienced pregnancy concurrently to being treated for RR-TB; (2) various care experiences, mostly negative as a result of the RR-TB diagnosis, and some positive; and (3) coping strategies that the women used.

I do believe that given how the current version of the article reads, the emphasis of the paper should be on ‘care experiences’ and the issue of dual identities should be addressed in relation to the care experiences. The rationale for this recommendation: exploring experiences to reveal the meaning they have for identity lends itself to a phenomenological study design. Given the emphasis in the paper on the management of RR-TB in pregnancy and the need to tailor care interventions for the specific needs of pregnant women with RR-TB, lends itself more to the grounded theory approach the authors stated they employed. Coping strategies would then need to be discussed in the light of person-centred models of care.

Specific comments/Recommendations

Given the qualitative approach to the study, I have used the COREQ checklist for reporting qualitative research (Tong, Sainsbury and Craig, 2007) in appraising this paper and in compiling this review.

1. Title: consider deleting ‘a qualitative study on’; consider inserting ‘care’; consider adding a context - to read: “Take the treatment and be brave”: Care experiences of pregnant women with rifampicin-resistant tuberculosis in Durban, South Africa.

Response: Thank you for the detailed comments and support for the paper. We appreciate them and the time and care taken with our work. We have noted throughout the paper that we are focused on the care experiences of the women. We have changed the title as you suggested, and it now reads ““Take the treatment and be brave”: Care experiences of pregnant women with rifampicin-resistant tuberculosis.”

2. Key words: consider adding ‘qualitative study’ or ‘qualitative research’.

Response: We have added “qualitative research” as a key word.

3. Abstract:

a. The background in the abstract is ambiguous. [“Little data on the management of pregnant women with RR-TB” could be understood to mean the clinical management of RR-TB. “… or their experiences” could mean the effects/side-effects of the clinical management. The background does not foreground the research problem that this paper seeks to address – the care experiences of women who are both pregnant and on treatment for RR-TB.

b. Methods – the purpose of the COREQ guidelines is to check the quality of reporting of studies employing qualitative methodologies, and do not guide purposive sampling. The qualitative study design/methodological orientation employed in the study should be stated in the methods. The word ‘documents’ is misleading, as the primary and only source of data was semi-structured interviews, which were then either transcribed (changed from oral data to textual data) or annotated. Iterativity usually refers to the cyclical nature of data collection and data analysis, and usually is employed to enrich data collection so that saturation of meaning/information power is achieved (see Hennink et al. 2016; Malterud et al 2015). In implementing a thematic network analysis, authors should state whether they adopted an inductive, deductive or hybrid approach in identifying the basic, organising and global themes.

c. Results: The presentation of results should align with the thematic network analysis adopted. Is “take the treatment and be brave’ the global theme? How do the organising themes illuminate the need to ‘be brave’? What does being brave ultimately mean for the care experience (note the singular versus the plural use) of pregnant women with RR-TB?

d. Conclusion: reads very generically and does not particularly align with the purpose and results of the study.

Response: We have made significant changes to the abstract as this reviewer suggested. We now note that the study is focused on the care experiences of the women, have been more specific in the methods and results section, and have noted that the experiences required the women to show sustained resilience and that is why the overarching theme is to “be brave”. 

4. Introduction: has a strong emphasis on the clinical aspects of TB and RR-TB in pregnancy and does not adequately bring under focus/foreground the need to explore the patient perspective/voice/experience of care. Further, the introduction may well pre-empt the findings in highlighting the dual identities. Unless the authors indeed acknowledge the dual identities upfront, and then focus the study on characterising the care experience given the dual identities.

Response: We have updated the introduction to include a paragraph on the importance of understanding the patient voice and experience in health care. We have also expanded the paragraph on dual identities in the care experiences of pregnant women with other serious medical problems. We have also added two new references in this section. 

5. Study design – the authors do not identify the methodological orientation underpinning the study. They do allude to a grounded theory approach to data analysis, but do not provide convincing evidence that they were indeed faithful to the methods of a grounded theory study design.

Response: We have added additional information to the methods section specifically on our use of grounded theory. We have also noted that the experiences we focused on analyzing were those that were part of their medical care. 

6. Study setting and population –

a. The authors identify the population that was studied as pregnant women receiving treatment for RR-TB. Usually qualitative designs are employed to study phenomena rather than populations. Phenomena are studied from the perspective of a particular group or groups.

Response: We agree that this was the study of a phenomenon—that of receiving medical care for both RR-TB and pregnancy—but we had to define the population both for the purposes of carrying out the study and for describing it according to the guidelines of the paper. Thus, we structured our methods section in this way. We did add the following sentence to the section on the study population: “The purpose of the study was to describe the phenomenon of receiving medical care for both pregnancy and RR-TB, and this required working with a population of women who were both pregnant and living with RR-TB.”

b. There is insufficient detail regarding the sampling: what purposive sampling strategy was employed? What was the rationale for the sampling strategy employed? What were the sampling criteria used to select the 17 pregnant women from the 108? Were all women pregnant at the time of recruitment? How were the respondents recruited?

Response: Upon further review of the data for this revision, it became clear that while there was an attempt to purposively sample women to be in this study, the participants were actually more of a convenience sample. We have updated the manuscript accordingly.

7. Data collection and analysis

a. What approach to data collection was used – general interview guide approach or was a standardised open-ended interview conducted? The interview schedule design and the ordering and framing of the questions suggest the latter approach, but in the analysis section the authors state that the interview guide was updated after each round of data collection. What updates were made?

Response: A general interview guide was used with the first participants and it was updated to include additional specific questions, which are included in the final guide submitted here. 

8. Results

a. The results skim over the ‘dual identities’ and quickly turn to the experiences of care – being given advice re: TOP.

b. I am not convinced that each of the themes relate to the phenomenon of dual identities. In my opinion most of what is reported relates to the care experience and speaks very loudly to the nature of care systems required – from compassionate health workers, to intersectoral support systems to care for their social needs, to health information, to integrated care systems, etc. I would recommend that the results be organise to characterise (in organising themes) the care experience.

c. I would recommend that the coping mechanisms identified be discussed in the light of how they can be harnessed to improve on the care for pregnant women with RR-TB, particularly within person-centred models of care.

d. The quotation – take the treatment and be brave – seems to be an encouragement, at minimum to be adherent to care processes, and at best to be full participants in the care process. It may well also speak to the need for peer mentoring/buddy system in the care process, similar to the Mothers to Mothers-to-Be (M2M2B) programme.

Response: Thank you for these comments on the results. We have edited them to try and focus more on the identities that arose as a result of the care experiences. We try to emphasize that it is the experience of receiving care within the health system that drives the development of the dual identities for the women. 

9. In the discussion I would opt to focus on the characterisation of the care experience and to even provide a visual diagram of the actions and interactions that influence the global characterisation of the care experience for pregnant women with RR-TB. The visual provided at present is labelled “Analytic framework” so it is not clear if this provided the framework for a deductive analysis or if this was the outcome of an inductive analysis. Generally grounded theory is used to generate an hypothesis, and based on the results, the hypothesis generated would need to identify the aspects that a care package for pregnant with RR-TB would need to address.

Response: We have noted that the analytic framework resulted from the inductive analysis. We agree with the reviewer that the study should identify aspects of a care package for pregnant women and have added this into Table 2 in the paper. 

10. Conclusion: does not seem to align with the purpose and nature of the study.

Response: We have edited the conclusion to try and align more with the purpose of the study: to describe the experiences of pregnant women with RR-TB and propose a package of care that might be offered to them to enhance their experiences. 

11. Limitations: I do not accept the limitation that the study cannot be generalizable to other populations. Firstly, you did not study a population – you studied a phenomenon (poorly defined but nevertheless can be named as ‘care experience of pregnant woman with RR-TB); secondly you implemented a qualitative study to study the phenomenon – qualitative studies do not aim to achieve generalizability – as you correctly stated you sampled information rich respondents to help to saturate the phenomenon. You did not sample to achieve representivity so that you can generalise to a population. In qualitative studies you aim for transferability – and you need to provide a rich context description so that your reader can assess from your description of the context how similar or dissimilar it is to their own, and whether the findings you report may be transferrable to their context. The real limitation is that you do not describe the context of care beyond the medical management of RR-TB. How does RR-TB get diagnosed in pregnant women, where are they assessed first, where are they referred to, what period of hospitalisation is required, what type of personnel cares for pregnant women with RR-TB at each level of care. This is all important background information that contextualises the care experience of pregnant women with RR-TB.

Response: Thank you for this comment. We have removed the sentence about generalizability from the limitations section and noted that it was not designed to achieve representivity. However, we did add a sentence about the limitation of using a convenience sample, which states: ‘It utilized a convenience sample and thus may not have captured a diverse range of experiences.”

12. As a final note: reflexivity is an important process in studies within the interpretivist paradigm and that generate qualitative data – how was this requirement met in this study?

Response: We have added this note on reflexivity in the discussion section: “Finally, and as part of the tradition of reflexivity that is essential in doing qualitative research, we note that two of us are engaged in providing care to people with RR-TB as medical providers and this may have impacted our understanding, analysis, and description of the experiences of the women who participated in this study. “

Reviewer #2: Thank you for such an important paper.

The manuscript highlights the experiences of pregnant women with RRTB. Overall, the pregnancy identity comes across a lot more than the patient identity. although the illness identity is discussed, the extent to which it is complicated by pregnancy does not come across strongly.

This was a study on pregnant women, the authors present the fears that the participants had regarding the health and development of their children (Line 508-509). It is not clear if the participants were re-interviewed after giving birth? or at any other point seeing that data collection was over a 5 year period. Perhaps clarify this under data collection.

Response: Thank you for raising this omission. To address the omission, we have revised line 181 to include ‘and during delivery.’ 

The authors point to the challenges faced by pregnant women at the ante-natal clinics, perhaps they could also make a suggestion that support be offered at this point and not only after delivery?

Response: Thank you for this suggestion. We have revised line 644 to include ‘ante- and post-partum periods.’ 

The authors also suggest (Line 587) counseling on how best to communicate with health care providers. This suggests a deficiency in the patients. The authors could rephrase this, perhaps to indicate the need for health workers to communicate better with patient, or for patients to assert themselves more, whatever the authors intend to put across.

Response: We have revised lines 644-647 to address the reviewers concern.

Thank you

Reviewer #3: This is a qualitative study of pregnant women who are receiving treatment for drug-resistant tuberculosis. Given the paucity of data on this particular group (all published data to date has only 100 pregnant women with DR-TB in total) and the use of qualitative methods, it is novel and worthy of sharing. The authors follow COREQ guidelines, and this is appreciated. I have some suggestions to the authors that I think , if addressed, will improve the manuscript. They are as follows:

MAJOR

(1) Abstract: The manuscript body nicely divides experiences as a pregnant woman and as a patient with TB into those that are contradictory (those in which a woman's role or responsibilities as mother and as MDR-TB patient contradict each other) and those that are overlapping (where actions taken to be the best soon-to-be mother and the best patient one are aligned). The abstract, however, fails to give examples of each of these, and one is left with generalities and statements that are somewhat self-evident. The conclusions fail to provide any concrete or specific ways we can help these women. I'd suggest a rewrite of the Results and Conclusions.

(2) Methods: Please describe how exactly participants were 'purposively selected'? Based on what criteria? To achieve what objectives?

Response: Upon further review of the data, it became clear that this sample was actually a convenience sample and not a purposive one and we have clarified this in the manuscript. 

(3) Results: Table 1 is not helpful. It merely lists the age, HIV status, and # of births (? including the present one?) of each participant. These statistics can be summarized in the main text.

Response: Thank you for your suggestion. We have deleted the table and as suggested included the data from the table in the text (lines 220-222).

(4) Results: Consider replacing the Figure with a Table that summarizes contradictory areas and overlapping areas, perhaps with some brief examples. This will be helpful to the reader, to help him/her organize and capture main ideas. I overall like the visual impact of this type of figure as an analytical framework, but here it doesn't work too well because the words in the left circle and the ones in the right circle are essentially the same, just reversed.

Response: Thank you for your suggestion. We have added Table 1 which summarizes contradictory areas and overlapping areas and provides brief examples. 

(5) Discussion: This would be enhanced if the authors would provide a prioritized list of ways care could be improved for pregnant women with DR-TB (lines 567-568 could be linked to a table that outlines recommended interventions)

Response: We have included a second table in the manuscript which details recommended elements of an optimized care package for pregnant women with RR-TB. 

MINOR

(1) The word 'data' is plural, so please make relevant adjustments (e.g. there are few data rather than there is little data)

Response: Thank you. We have corrected this through the manuscript.

(2) Lines 120-122- what type of discrimination, and what are the reasons for this discrimination?

Response: To clarity the type of discrimination and reasons for it, we have revised lines 141-142 by adding “This discrimination, based on a fear of RR-TB transmission, can result in sub-optimal care and isolation during delivery and the post-partum period and……” 

(3) Lines 124-125- give a short example of how identities of prospective mother and patient with HIV or cancer are at odds so the reader knows what you mean here. You explain it later in the Discussion, but it would be better to introduce this 'contradictory roles' idea early in the paper.

Response: To clarify the dual identity phenomenon we have revised lines 146-152 to include a description of the dual identity phenomenon experienced by a pregnant woman with cancer. 

(4) Lines 133-124- it took 6 years to enroll this 17-person study. Why?

Response: Thank you for alerting us to this error. This qualitative study was nested in an observational cohort study for which we recruited women over 6 years. However, for this qualitative study we conducted interviews over a two-year period only. This has been corrected in the text (lines 133-134).

 (5) Avoid inflammatory or hyperbolic language (e.g. pressured instead of harassed, vulnerable instead of incredibly vulnerable...); the point is made without that.

Response: Thank you for alerting us to our use of hyperbolic language. We have edited the whole manuscript and removed inflammatory words.

(6) Consider putting findings related to termination of pregnancy and also about mandatory hospitalization into the abstract so they get more visibility. These are really important issues and are concrete and illustrative examples of challenges these women face when having the dual role of pregnant woman and patient with TB

(7) How did these 17 women do?

Response: The outcomes of these women are reported in another study (Loveday et al., 2020, ClD, https://academic.oup.com/cid/advance-article/doi/10.1093/cid/ciaa189/5788430), and since we are reporting on their treatment experiences here, we opted not to include them. 

(8) Did the research team organize support groups for these women? It seems that such a group, with an advocate participating, would be valuable.

Response: We did not organise formal support groups but support and counselling was provided to these women as part of standard of care. In our table of recommendations, we did add peer support, including support groups. 

(9) Lines 397-398: would assume that women were not always in control of their reproductive health

Response: This was not an assumption but rather something that was expressed by some of the women in the study. In the communities from which many of these women came/were living, there are high rates of gender-based violence. At the same time, women are often valued and defined by their roles as mothers. When these conditions are coupled with lack of easy access to contraception, it is not surprising that women who participated in this study expressed feelings that they were not always in control of their reproductive health. 

(10) The authors make a point of stating that it was unexpected that the women's husbands/ partners would play a role in supporting them during this difficult time. Consider describing why this was unexpected.

Response: Thank you for raising this issue. To clarify, we have added the following to the text (lines 492 – 494) “In our setting, patriarchy is very common, and the care of children and the sick considered the responsibility of women.”

---

## [Decision Letter · Decision Letter 1]

3 Nov 2020

PONE-D-20-13335R1

"Take the treatment and be brave": Care experiences of pregnant women with rifampicin-resistant tuberculosis

PLOS ONE

Dear Dr. Loveday,

Thank you for submitting your manuscript to PLOS ONE. Both rveiewers of the revised manuscript felt that the comments were addressed, however one of the reviewers has suggestions for minor edits, and I agree that these will improve the manuscript.  Once addressed, the paper can be accepted without further review.

We look forward to receiving your revised manuscript.

Kind regards,

Jennifer Zelnick

Academic Editor

PLOS ONE

Additional Editor Comments (if provided):

Please see very minor comments from reviewer #3.

Reviewers' comments:

Reviewer's Responses to Questions

**Comments to the Author**

1. If the authors have adequately addressed your comments raised in a previous round of review and you feel that this manuscript is now acceptable for publication, you may indicate that here to bypass the “Comments to the Author” section, enter your conflict of interest statement in the “Confidential to Editor” section, and submit your "Accept" recommendation.

Reviewer #1: All comments have been addressed

Reviewer #3: All comments have been addressed

2. Is the manuscript technically sound, and do the data support the conclusions?

Reviewer #1: Yes

Reviewer #3: Yes

3. Has the statistical analysis been performed appropriately and rigorously? 

Reviewer #1: N/A

Reviewer #3: Yes

4. Have the authors made all data underlying the findings in their manuscript fully available?

Reviewer #1: No

Reviewer #3: Yes

5. Is the manuscript presented in an intelligible fashion and written in standard English?

Reviewer #1: Yes

Reviewer #3: Yes

6. Review Comments to the Author

Reviewer #1: Thank you for asking me to review the resubmission of the paper entitled: “Take the treatment and be brave”: Care experiences of pregnant women with rifampicin-resistant tuberculosis. (PONE-D-20-13335)

1. Abstract – revised satisfactorily. But please note the following:

a. The purpose of the COREQ guidelines is to check the quality of reporting of studies employing qualitative methodologies. The COREQ guidelines do not guide sampling strategy.

b. Please correct typo: “… 3) experience of the care they received for their pregnancy and their RR-TB; and 4) …”

2. Introduction – revised satisfactorily.

3. Methods and materials – revised satisfactorily. Suggested edits as follows:

a. Line 157: “This was a qualitative study generating data using open-ended interviews …”

b. Line 166: “The qualitative study was part of a larger cohort study of pregnant women …”

c. Line 169: “A convenience sample of 17 women was selected …”

d. Line 173: “The sample of 17 women participated in…”

4. Results – revisions very acceptable!

5. Discussion – Revised satisfactorily. Suggested edits:

a. Generally I would recommend removing the definitive article ‘the’ when referring to ‘the women’. While you did indeed interview a specific group of respondents, and your findings are based on the data generated by these women, you are discussing a phenomenon, and your discussion should be transferrable to pregnant women in similar contexts; i.e. the discussion needs to be of the broader phenomenon of ‘care experiences of pregnant women with RR-TB’; and not of the women in this study (as you have done in the conclusion).

6. Limitations – Revised satisfactorily.

Reflexivity – statement that has been inserted is acceptable. However, I would recommend the authors read more extensively on the issues of reflexivity (epistemological, methodological, axiological, critical) and engage with the concept throughout the research process in future studies.

Reviewer #3: The authors have revised the manuscript to respond to this reviewer’s comments. Responses are adequate and I have no further comments.

7. PLOS authors have the option to publish the peer review history of their article (what does this mean?). If published, this will include your full peer review and any attached files.

Reviewer #1: **Yes: **Anna Voce

Reviewer #3: No

---

## [Author Response · Author response to Decision Letter 1]

4 Nov 2020

RESPONSES TO REVIEWERS

PONE-D-20-13335R1

"Take the treatment and be brave": Care experiences of pregnant women with rifampicin-resistant tuberculosis

PLOS ONE

Reviewer #1: 

Reviewer #1 notes that we have not made our data available. We have made our data available. 

1. Abstract – revised satisfactorily. But please note the following:

a. The purpose of the COREQ guidelines is to check the quality of reporting of studies employing qualitative methodologies. The COREQ guidelines do not guide sampling strategy.

b. Please correct typo: “… 3) experience of the care they received for their pregnancy and their RR-TB; and 4) …”

Response: Thank you for raising these concerns. We have addressed these in the revised version of the manuscript which we have submitted.

2. Introduction – revised satisfactorily.

3. Methods and materials – revised satisfactorily. Suggested edits as follows:

a. Line 157: “This was a qualitative study generating data using open-ended interviews …”

b. Line 166: “The qualitative study was part of a larger cohort study of pregnant women …”

c. Line 169: “A convenience sample of 17 women was selected …”

d. Line 173: “The sample of 17 women participated in…”

Response: Thank you for these suggested corrections. We have revised the manuscript accordingly. 

4. Results – revisions very acceptable!

5. Discussion – Revised satisfactorily. Suggested edits:

a. Generally I would recommend removing the definitive article ‘the’ when referring to ‘the women’. While you did indeed interview a specific group of respondents, and your findings are based on the data generated by these women, you are discussing a phenomenon, and your discussion should be transferrable to pregnant women in similar contexts; i.e. the discussion needs to be of the broader phenomenon of ‘care experiences of pregnant women with RR-TB’; and not of the women in this study (as you have done in the conclusion).

Response: Thank you for this suggestion. By removing ‘the’ together with some other minor edits the discussion is now more transferrable to pregnant women in similar contexts. 

6. Limitations – Revised satisfactorily.

Reflexivity – statement that has been inserted is acceptable. However, I would recommend the authors read more extensively on the issues of reflexivity (epistemological, methodological, axiological, critical) and engage with the concept throughout the research process in future studies.

Response: Thank you for this suggestion. We will read more extensively on the issues of reflexivity for future studies.

---

## [Editor Report · Decision Letter 2]

6 Nov 2020

"Take the treatment and be brave": Care experiences of pregnant women with rifampicin-resistant tuberculosis

PONE-D-20-13335R2

Dear Dr. Loveday,

We’re pleased to inform you that your manuscript has been judged scientifically suitable for publication and will be formally accepted for publication once it meets all outstanding technical requirements.

Kind regards,

Jennifer Zelnick

Academic Editor

PLOS ONE

Additional Editor Comments (optional):

Thanks for your quick response and revisions to the manuscript.
---

## [Editor Report · Acceptance letter]

10 Dec 2020

PONE-D-20-13335R2 

*“Take the treatment and be brave”:* Care experiences of pregnant women with rifampicin-resistant tuberculosis 

Dear Dr. Loveday:

I'm pleased to inform you that your manuscript has been deemed suitable for publication in PLOS ONE. Congratulations! Your manuscript is now with our production department. 

Kind regards, 

on behalf of

Dr. Jennifer Zelnick 

Academic Editor

PLOS ONE